# Transcriptome and Metabolome Analyses of Leaves from Cutting Rejuvenation of Ancient *Cinnamomum camphora*

**DOI:** 10.3390/ijms25147664

**Published:** 2024-07-12

**Authors:** Lipan Liu, Aihong Yang, Tengyun Liu, Shujuan Liu, Ping Hu, Caihui Chen, Hua Zhou, Jingfang Wu, Faxin Yu

**Affiliations:** Jiangxi Provincial Key Laboratory of Improved Variety Breeding and Efficient Utilization of Native Tree Species (NO.2024SSY04092), Institute of Biological Resources, Jiangxi Academy of Sciences, No. 7777, Changdong Road, Nanchang 330096, China; liulipan@jxas.ac.cn (L.L.); yangaihong@jxas.ac.cn (A.Y.); liutengyun@jxas.ac.cn (T.L.); liushujuan@jxas.ac.cn (S.L.); huping@jxas.ac.cn (P.H.); chencaihui@jxas.ac.cn (C.C.); zhouhua@jxas.ac.cn (H.Z.); wujingfang@jxas.ac.cn (J.W.)

**Keywords:** ancient *Cinnamomum camphora*, rejuvenation, phytohormone, primary metabolites, transcriptome

## Abstract

Rejuvenation refers to the transition from the state of mature to juvenile. Many ancient *Cinnamomum camphora* have aged and died due to climatic and anthropic factors. Vegetative propagation can protect valuable germplasm resources. In this study, a 2000-year-old ancient *C. camphora* and its 2-year-old cutting plantlets were selected as experimental materials. The results indicated that the number of leaves with palisade tissue (Pal) cell layers was different between samples, with two layers in the rejuvenated leaves (RLs) and one layer in the mature leaves (MLs) and young leaves (YLs). Indole-3-acetic acid (IAA), isopentenyladenine (iP) and isopentenyladenosine (iPR) concentrations were significantly higher in RLs than in MLs and YLs, but the abscisic acid (ABA) concentration was lower. Targeted metabolome analysis identified 293 differentially accumulated metabolites (DAMs). Meanwhile, a total of 5241 differentially expressed genes (DEGs) were identified by transcriptome sequencing. According to the KEGG analysis, there were seven important enriched pathways in the MLs, RLs and YLs, including plant hormone signal transduction (57 DEGs), plant–pathogen interaction (56 DEGs) and MAPK signaling pathway–plant (36 DEGs). KEGG enrichment conjoint analyses of DEGs and DAMs identified 16 common pathways. Integrated analyses of cytological, hormone, metabolome and transcriptome elements can provide a research basis in regard to the rejuvenation regulatory mechanism of ancient *C. camphora*.

## 1. Introduction

Rejuvenation refers to the whole or partial reversal of mature plant characteristics during the mature to juvenile development stage, regressing to a previous stage of juvenile development and regaining part or all of the growth characteristics of young plants [1]. The growth and development of trees include juvenile and mature stages, which have differences in terms of phenotype, structure, and physiology [2,3,4,5]. New stems can resprout naturally from the trunk or root of old *Ginkgo biloba* trees. The leaf thickness, lobe number, fresh weight, and water content are dramatically increased in resprouters compared to old branches [6]. The leaves have significant changes in morphological, cytological, and physiological characteristics under different truncation rejuvenation treatments [7]. Ancient trees do not exist as independent individuals and are a valuable resource in the population [8]. Ecologically, ancient trees are anchored nodes of biodiversity and ecosystem complexity, and they have irreplaceable functions in the whole ecosystem restoration [9]. Moreover, ancient trees are a symbol of long history and culture, and they have very high scientific, cultural, landscape and economic values [10]. The gene map of ancient trees contains valuable genes, including genes linked to anti-ageing, pest resistance and disease resistance, which are important materials for plant genetic improvement. Ancient trees refer to trees that are more than one hundred years old and are in a declining period of vegetative growth or even on the verge of death. Asexual propagation is considered the only way to preserve and utilize forest germplasm resources, because it can maintain the genetic background stability of the mother tree. Vegetative propagation is an important method of asexual reproduction and rejuvenation in forest plants. Because cutting has the characteristics of simple technology, convenient sampling, a short seedling cycle, and a large propagation scale, it has become the most widespread method of vegetative propagation for preserving forest germplasm resources, especially those of ancient trees [11]. However, the changes that occur to the juvenile characteristics through cutting have not been characterized in regard to ancient *C. camphora* trees.

Plant hormones have been identified trace substances which are essential for plant growth and development. Plant hormones, including cytokinin, ethylene, auxin, gibberellin (GA), abscisic acid (ABA), salicylic acid (SA) and jasmonic acid (JA) play important regulatory roles in plant leaf senescence and rejuvenation [12]. Phytohormones not only function independently but also interact with each other and are regulated by feedback [13]. The content changes in hormones can reflect juvenile and senescent plant levels [1]. The endogenous phytohormone contents of indole-3-acetic acid (IAA), GA3, and zeatin gradually increased, but ABA decreased in leaves with a serial subculture of in vitro apple shoot rejuvenation [14]. The leaf endogenous hormone contents of IAA, zeatin riboside (ZR), GA3, and ABA changed through trunk truncation rejuvenation in ginkgo trees [7]. Moreover, phytohormone treatment can induce tissue or organ regeneration and produce rejuvenated plants. Cytokinins and GA_4_ phytohormones can promote dedifferentiation and rejuvenation in *Nicotiana tabacum* tissue culture [15].

The multi-omics analysis of transcriptomic, metabolomic, proteomic and epigenomic data has gradually become a research hotspot along with the rapid development of biological information technology. Plant senescence and rejuvenation are complex biological processes which are regulated by interactions among multiple genes. In *Eucalyptus grandis*, the floral gene *EgSTK* was expressed at low levels in young leaves (YLs) but was not expressed in mature leaves (MLs) [16]. The phenylpropanoid, naringenin, lignin, cutin, suberin, and wax biosynthesis KEGG pathways were significantly enriched in the MLs of *Toona sinensis* [17]. Integrated analyses of the transcriptome and metabolome of YLs and MLs identified 1256 differentially expressed genes (DEGs) and 38 differentially accumulated metabolites (DAMs) in ginkgo trees [18]. Analyses of the YL and ML transcriptome identified 2082 DEGs and 275 DAMs in *Yunnanopilia longistaminea* [19]. The development of forest tissue is dynamic in time and space. Transcriptome analysis of different tissues revealed that most isoflavone biosynthetic genes were lowly expressed in the YLs and MLs of *P. candollei var. mirifica*, and *PmMYB24*, *PmMYB77*, *PmCYP81Es*, *PmIFRs*, and *PmPT* were highly expressed in MLs [20]. A total of 501 common DEGs were identified in different tissues and the developmental stages of sweet potato (YLs, MLs, young stems, mature stems, and storage roots), and they were mainly enriched in secondary metabolite biosynthesis [21]. The flavonoid compound contents were different among different tea plant cultivars and tissue materials (including YLs, MLs, old leaves, young roots and stems), which were related to 11 gene-encoding biosynthetic enzymes that exhibited different expression patterns [22].

The cell structure, metabolism and gene expression observed simultaneously occurring changes during the leaf senescence process [23]. The differentially expressed transcription factors (TFs) bHLH, MYB, ERF, MYB-related, NAC, and WRKY play important regulatory roles during seasonal leaf senescence in *Populus trichocarpa* [24]. The conjoint analysis of transcriptome and metabolome data is an important approach to excavating functional genes and metabolites in plant growth and development. The majority candidate genes and metabolites of flavonoid biosynthesis pathway were down-regulated during the leaf development process [25]. During the transition from mature to juvenile growth stage, plant rejuvenation can regain juvenile traits and activities through transcriptional regulatory changes [26]. Transcriptome analysis revealed expressed genes involved in photosynthetic capacity, auxin-signaling pathway, and stress-associated hormones were up-regulated in the leaves of resprouters than old branches in ginkgo trees [6]. Trunk truncation is an important rejuvenation method involved in the vegetative growth phase transition in *G. biloba*, and it can alter endogenous hormone levels and effectively increase leaf biomass and flavonoid accumulation [7]. Through comparative transcriptome and gene function identification profiles in the RLs and MLs of *Robinia pseudoacacia*, researchers identified the *RpTOE1-RpFT* module associated with rejuvenation during vegetative propagation, through which *RpTOE1* can directly bind to the promoter of *RpFT* [27].

The whole process of plant growth and development is affected by the external environment and various internal factors. Plant leaves are the first barrier against the external environment and play an important biological role in plant resistance to biotic and abiotic stresses. The photosynthetic activity of YLs was more susceptible and vulnerable than MLs under cadmium (Cd) stress in soybean plants [28]. A total of 67.8% DEGs were identified during the transition from YLs to MLs, and DEGs involved in cyanogenic metabolism, lignin and anthocyanin biosynthesis correlated with the change in the disease resistance of leaves [29].Targeted metabolome and transcriptome analysis revealed that the up-regulated genes were mainly enriched in stress tolerance, signal transduction and secondary metabolite biosynthesis pathways in old leaves than in the fruit and new leaves of *Olea europaea* [30]. The physiological metabolism and gene expression of YLs and MLs showed significant differences under Fe-deficiency stress in *Gynura bicolor* [31].

*Cinnamomum camphora* (Linn.) Presl, as a member of the genus *Cinnamomum* (Lauraceae) and belonging to an evergreen broad-leaved tree species, has been regarded a representative tree species in tropical and sub-tropical regions [32]. *C. camphora* is widely distributed south of the Yantze River in China, and the largest population of this type of tree is located in the Jiangxi Province [33]. This species has important economic, ornamental, cultural, and ecological value, and it is used in timber, flavors, medicine, and chemical products [34]. The genome sequence of *C. camphora* was published in November 2021, and this information will greatly promote the development of functional genome research on the camphor tree [35]. An ancient *C. camphora* tree is classed as one aged more than 100 years. They contain important germplasm resource information and are the material foundation of scientific research and application. Ancient *C. camphora* trees are widely distributed in the countryside and urban areas of China, and the wild population has become a local tourist landscape. The ancient *C. camphora* species was cultivated by ancestors of different historical dynasties and has a longer history of exploitation and utilization, and the earliest cultivation period can be traced to 2000 years ago [33]. Currently, ancient *C. camphora* trees are involved in symbols and signs in traditional Chinese culture, including heritage trees, sacred trees, and city trees [36]. Due to natural and human factors, wild populations of ancient *C. camphora* have suffered serious destruction, and they are now listed on the “China Species Red List” [37]. However, most ancient *C. camphora* trees have been explored only in situ conservation situations at present, and the development of ex situ conservation is extremely urgent in terms of asexual reproduction.

In this study, we selected a 2000-year-old ancient *C. camphora* and its 2-year-old cutting plantlets as the experimental materials. The leaf cytological characteristics were observed. Ultra high-performance liquid chromatography-tandem mass spectrometery (UPLC-MS/MS) was used to detect plant hormone and primary metabolite contents. In addition, DEGs were identified through transcriptome analysis, and GO and KEGG enrichment analyses were also carried out. The RNA-seq data were verified by qRT–PCR analysis. Finally, the conjoint analyses of DEGs and DAMs were carried out. The integrated analyses of cytological, hormone, metabolomic and transcriptomic data can provide a research basis for the rejuvenation regulatory mechanism of ancient *C. camphora*.

## 2. Results

### 2.1. The Leaf Phenotype and Tissue Structure of Ancient C. camphora and Its Cutting Plantlets

The leaf phenotype and tissue structure of MLs, RLs, and YLs are shown in Figure 1. The leaf area was greatest in the MLs, and the upper epidermal color was the darkest. The RLs were smaller and more tender than the MLs and YLs. As the most prominent structural feature, the palisade tissue (Pal) cells were closely arranged in two layers in the RLs, but the spongy tissue (Spo) cells were loosely arranged in one layer in the MLs and YLs. In the MLs, the vessel (Ve) diameter and number were greater than in the RLs and YLs. The upper epidermis (Uep) cells were regularly arranged in the MLs and YLs, but loosely arranged in the RL. The lower epidermis (Lep) contained some stomata (St) on the MLs and YLs but rarely on the RLs. Cuticle (Cu) thickness was obviously greater in the MLs and YLs than in the RLs.

### 2.2. Analysis of Plant Hormone Contents in the Leaves of an Ancient C. camphora and Its Cutting Plantlets

Based on the UPLC–MSMS platform (1290 Infinity UPLC–5500 QTRAP), the concentrations of 12 plant hormones were measured, and the concentrations are shown in Figure 2. The IAA, isopentenyladenine (iP) and isopentenyladenosine (iPR) concentrations of RLs were significantly higher than MLs and YLs (*p* < 0.01). However, the ABA concentration of MLs and YLs was higher than RLs (*p* < 0.05). The SA and JA concentrations were lower in RLs than MLs (*p* < 0.05). The trans-zeatin (tZ) and cis-zeatin (cZ) concentrations were higher in RLs than MLs and YLs, but there was no significant difference. In addition, the trans-zeatin riboside (tzR) and cis-zeatin riboside (czR) concentrations were significantly lower in RLs than MLs and YLs (*p* < 0.01).

### 2.3. Identification of DAMs in the Leaves of Ancient C. camphora and Its Cutting Plantlets

In this result, the primary metabolites were measured using a targeted metabolome. Principal component analysis (PCA) showed each of three samples of biological replicates clustered together (Figure 3A), indicating that the metabolite data could be used for subsequent analysis. A total of 726 primary metabolites were detected via a UPLC–MS/MS platform and a self-built database, and the total primary metabolite information is listed in Appendix A. These metabolites were divided into four classes (Figure 3B): amino acids and derivatives (32.64%), lipids (26.03%), organic acids (17.36%), nucleotides and derivatives (9.37%), and others (14.6%). A total of 293 DAMs were identified based on variable importance in projection (VIP) > 1, *p* < 0.05, and |log_2_(Fold Change)| > 1. The numbers of DAMs were 235 in the ML vs. RL group, 89 of which were down-regulated and 146 of which were up-regulated (Table 1; Appendix A). The numbers of DAMs were 111 in the YL vs. RL group, 62 of which were down-regulated and 49 of which were up-regulated (Table 1; Appendix A). The numbers of DAMs were 92 in the ML vs. YL group, 38 of which were down-regulated and 54 of which were up-regulated (Table 1; Appendix A). There were 15 common DAMs in the Venn diagram analysis (Figure 3C). A heatmap of the DAMs cluster analysis is shown in Appendix A, and the up- and down-regulated DAMs are clustered together.

Annotation results of DAMs were classified according to a KEGG pathway type, as shown in Appendix A. Metabolic pathways contained the largest percentage of DAMs in the comparison analyses (ML vs. RL, YL vs. RL, and ML vs. YL). Enrichment analysis of the DAMs in the KEGG pathway was conducted, as shown in Figure 4, in which the rich factor was loosely positively related to the greater enrichment degree. The top 20 pathways ranked by *p* value are shown in Figure 4. In the ML vs. RL comparison, the DAMs were significantly enriched in the biosynthesis of various plant secondary metabolites and the biosynthesis of cofactor pathways (*p* < 0.05). In the YL vs. RL comparison, the DAMs were significantly enriched in the C5-branched dibasic acid metabolism and sulfur metabolism pathways (*p* < 0.05). In the ML vs. YL comparison, there was no significant enrichment of DAMs in any pathway (*p* < 0.05).

### 2.4. DEG Identification and Analysis in the Leaves of an Ancient C. camphora and Its Cutting Plantlets

Based on Illumina NovaSeq platform, a total of 63.98 Gb of raw data were produced, and 59.92 Gb of clean data were obtained after quality control. The RNA-seq yielded 396.97 million clean reads (40.64–48.54) from three sample alignments to the *C. camphora* reference genome (GWHBGBX00000000) [35]. The sequence data useful ratio ranged from 93.38 to 93.85%, and the Q30 values were above 94.15%. A total of 10,019 genes were identified, and the total gene information is listed in Appendix A. The alignment ratio was 92.21–93.13% of the clean data mapped to the reference genome (Appendix A).

The PCA results showed that the YL, ML, and RL samples were clearly separated, so the sequence data and biological replicates could be used for subsequent analysis (Figure 5A). The gene annotation summary of RNA-seq data mapped to different databases is shown in Appendix A. A total of 5241 DEGs were screened based on the criteria of |log_2_(Fold Change)| > 1 and *p* < 0.05 (Figure 5B, Appendix A). The ML vs. RL comparison identified 4194 DEGs (1825 up-regulated and 2369 down-regulated). A total of 2635 DEGs (1127 up-regulated and 1508 down-regulated) were identified in the YL vs. RL comparison. The ML vs. YL comparison identified 1988 DEGs (922 up-regulated and 1066 down-regulated). The volcano plots of the DEGs are shown in Figure 6A–C for the comparative analysis of MLs, RLs, and YLs. Clustering analysis of 5241 DEGs is shown in Figure 5C between the YLs, MLs, and RLs. A Venn diagram of DEGs revealed 429 common DEGs between the comparative analyses of MLs, RLs, and YLs (Figure 5D).

The GO enrichment process included three biological terms: cellular component (CC), biological process (BP), and molecular function (MF) of the DEGs. The cell periphery, plasma membrane, and membrane functions were considered important processes (Appendix A). The top 20 enriched KEGG pathways were identified from the KEGG analysis of MLs, RLs, and YLs, and the results are shown in Figure 6D–F. The top 20 enriched KEGG pathways included the photosynthesis, starch and sucrose metabolism, plant hormone signal transduction, plant−pathogen interaction, MAPK signaling pathway–plant, and phenylpropanoid biosynthesis pathways, among others. According to the KEGG pathway analysis, seven important pathways, including the flavonoid biosynthesis, isoquinoline alkaloid biosynthesis, MAPK signaling pathway–plant, phenylpropanoid biosynthesis, plant hormone signal transduction, plant–pathogen interaction, and starch and sucrose metabolism pathways, were identified in the comparative analyses of MLs, RLs, and YLs.

In the plant hormone signal transduction pathway (ko04075), we identified 57 DEGs. Compared to MLs and YLs, 19 DEGs were up-regulated in RLs (Figure 7A), such as AUX1 (*Ccam04G001031*), GH3 (*Ccam05G000465*), SAUR (*Ccam03G000841*), bAHP (*Ccam01G000736*), B-ARR (*Ccam02G000363*), BKI1 (*Ccam08G000264*), BSK (*Ccam09G000775*), CYCD3 (*Ccam02G002718*). In contrast, 17 DEGs were up-regulated in MLs compared to RLs and YLs, such as TF (*Ccam07G001230*), SIMKK (*Ccam02G000155*), EBF1/2 (*Ccam03G002668*), JAZ (*Ccam03G001140*), MYC2 (*Ccam06G002000*, *Ccam06G000478*), TGA (*Ccam04G000983*, *Ccam01G000959*, *Ccam01G002236*), and PR-1 (*Ccam01G002394*, *Ccam01G002397*).

A total of 56 DEGs were identified related to the plant–pathogen interaction pathway (ko04626). Notably, 23 DEGs were up-regulated in MLs compared to RLs and YLs, such as WRKY33 (*Ccam03G000614*, *Ccam03G003137*), MKK4/5 (*Ccam02G000155*), and PR1 (*Ccam01G002394*, *Ccam01G002397*) (Figure 7B). In contrast, 17 DEGs were up-regulated in RLs compared to MLs and YLs, such as CNGCs (*Ccam03G003435*, *Ccam04G001244*) and RbohD (*Ccam03G001119*).

A total 36 DEGs related to the MAPK signaling pathway–plant (ko04016) were identified. There were 15 up-regulated DEGs in MLs, including WRKY33 (*Ccam03G000614*, *Ccam03G003137*), MKK4/5 (*Ccam02G000155*), PR1 (*Ccam01G002397*, *Ccam01G002394*), MKK9 (*Ccam01G003582*), MYC2 (*Ccam06G002000*, *Ccam06G000478*), and CAT1 (*Ccam09G000181*, *Ccam03G000147*) (Figure 7C). In contrast, 13 DEGs were up-regulated in RLs, including ETR/ERS (*Ccam05G002716*, *Ccam09G001692*, *Ccam04G003069*), ERF1 (*Ccam03G001071*), RAN1 (*Ccam01G000405*), and RbohD (*Ccam03G001119*).

In the photosynthesis (ko00195) pathway, 26 genes were down-regulated, and only two genes, the photosystem II gene PsbQ (encoding photosystem II oxygen-evolving enhancer protein 3, *Ccam09G000268*) and f-type ATPase gene alpha (encoding F-type H^+^/Na^+^-transporting ATPase subunit alpha, *Ccam05G001695*), were significantly up-regulated in RLs compared to MLs and YLs (Figure 7D). The expression levels of all 10 light-harvesting chlorophyll−protein complex (LHC) genes were down-regulated in the photosynthesis-antenna protein pathway (ko00196) in RLs compared to MLs and YLs (Figure 7E).

### 2.5. The Sequence Data Conjoint Analysis of DEGs and DAMs

In order to explore the relationship of genes and metabolites, KEGG enrichment conjoint analyses of DEGs and DAMs showed there were 53 common pathways in the ML vs. RL comparison, 38 common pathways in the YL vs. RL comparison, and 26 common pathways in the ML vs. YL comparison (Figure 8A–C). There have been 16 common enriched KEGG pathways. The plant hormone signal transduction; glutathione metabolism; linoleic acid metabolism; isoquinoline alkaloid biosynthesis; and the tropane, piperidine, and pyridine alkaloid biosynthesis pathways were significantly enriched (*p* value gene < 0.01). The common numbers of DEGs and DAMs were identified via KEGG enrichment conjoint analyses, and the results are shown in Appendix A. Then, the DEGs and DAMs of Pearson correlation coefficients (PCCs) were calculated in different comparative groups. The nine quadrant graphs reveal the regulatory relationship of DEGs and DAMs (Figure 8D–F). The expression patterns of DEGs and DAMs were consistent in the third and seventh quadrants, indicating that the gene may be positively regulated metabolite. However, the DEGs and DAMs patterns from the first and ninth quadrants were opposite from each other, suggesting gene and metabolite expression is negatively correlated. We further selected the DAMs with PCCs greater than 0.8 and constructed correlation coefficient cluster heatmaps (Figure 8G–I). The results showed that a large number of DEGs were positively or negatively correlated with DAMs.

### 2.6. qRT–PCR Verification of DEGs

In order to verify the RNA-seq data of MLs, RLs, and YL, we selected eight genes to determine relative expression levels by qRT-PCR analysis, and they were gathered from the significant DEGs (Ccam01G000281, Ccam05G002712, Ccam09G001101, and Ccam09G001102) and important enriched pathways (plant hormone signal transduction: Ccam03G001071; MAPK signaling pathway–plant: Ccam03G001119 and Ccam09G000012; plant–pathogen interaction: Ccam02G001584). The results showed that the relative expression levels of qRT-PCR had a similar pattern trend with the FPKM values of RNA-seq data (Figure 9). Therefore, the transcriptome data were reliable and could be used for subsequent analysis.

## 3. Discussion

### 3.1. Leaf Differences between Juvenile and Mature Plants

Ancient trees have evolved to exhibit exceptional longevity, even living for hundreds to thousands of years [38]. The preservation of precious species resources is extremely urgent due to the ageing and death of ancient trees. The ancient *C. camphora* species has survived for hundreds of years, or even thousands of years, and its physiological age is in the decline stage of growth and development. Many ancient *C. camphora* trees have aged and died, and precious germplasm resources have been gradually lost. Vegetative propagation can maintain the uniform character and genotype of the parent plants, so it has been widely used for forestry production and application. In recent clonal forestry development, many propagation methods, including cutting, tissue culture, grafting, micropropagation, cropping and pruning, have been applied to produce juvenile plants from mature plants [39]. Numerous studies have reported changes in phenotypic, physiological, and anatomical structure in the transition from mature to juvenile plants, such as leaf shape and size [6], leaf cell ultrastructure [40], and leaf morphological and physiological characteristics [26]. Generally, the leaf phenotypic index can be used to assess the rejuvenation effect [13]. In our study, the leaf area of RLs was smaller than MLs from ancient *C. camphora* trees. The results were consistent with a previous report showing the leaf area and leaf length/width ratio were lower in juvenile black locust trees than in mature trees [40]. Moreover, Pal cells were closely arranged in two layers in the RLs, but the Spo cells were loosely arranged in one layer in the MLs and YLs. The same leaves anatomical were observed in three different ages of healthy *Platycladus orientalis*, and the Pal and Spo cell thicknesses were highest in young trees and lowest in old trees [41]. Therefore, the leaf area and Pal cell layers may be related to the transition from mature to juvenile in ancient *C. camphora*.

### 3.2. Plant Hormone Contents and Gene Expression Regulatory Function

Plant hormones are trace organic substances synthesized in plants that have significant effects on plant growth and development. Endogenous phytohormones play an important role in tree rejuvenation [42]. Auxins, cytokinins, and GAs can maintain juvenile characteristics following rejuvenation [7,14,43]. Phytohormone content analysis revealed that most hormones exhibited differences, and the content of the IAA in new shoots was higher than that in MLs of tea (*Camellia sinensis*) cuttings among three cultivars with different rooting abilities [44]. The balance of endogenous phytohormones is crucial for plant senescence and rejuvenation. The IAA/ABA ratio was closely related to in vitro juvenile phenotype and the rooting ability of tender stems [14,45]. The cytokinin level can be considered a reliable biochemical marker for evaluating the forest rejuvenation effect [46]. In our study, the phytohormone concentrations of IAA, iP, and iPR were significantly higher in RLs than MLs and YLs (*p* < 0.01), but the ABA concentration was higher in MLs and YLs than RLs, oppositely (*p* < 0.05). The results of this study are consistent with those of the above-mentioned reports. Therefore, the IAA, iP, and iPR have positive functions, but ABA has the opposite function in the rejuvenated cutting of ancient *C. camphora* trees.

During plant development, hormones can mediate cell division and growth to induce tissue differentiation and regeneration, further promoting rejuvenation of mature plants [42]. The hormone signal transduction pathway plays an important role in the juvenile state being maintained on a transcriptional level [6]. The hormone signal transduction, photosynthesis and flavonoid biosynthesis pathways play important roles in flavonoid biosynthesis in ginkgo leaves of different ages [47]. Most genes were enriched in the plant hormone signal transduction pathway in the new shoots and MLs of tea cuttings [44]. In our study, the KEGG functional enrichment analysis of DEGs revealed that the plant hormone signal transduction pathway was commonly enriched in the comparative analysis of MLs, RLs, and YLs. A total of 57 DEGs, including AUX1, GH3, SAUR, bAHP, B-ARR, BKI1, BSK, CYCD3, SIMKK, EBF1/2, JAZ, MYC2, TGA and PR-1, related to the plant hormone signal transduction pathway (ko04075) were identified. Furthermore, through the KEGG enrichment conjoint analysis of DEGs and DAMs between the comparative analysis of MLs, RLs, and YLs, the plant hormone signal transduction pathway was important in the comparative analysis.

### 3.3. Identification of Differential Accumulated Primary Metabolites

Plant rejuvenation refers to the process of reversing a plant’s mature stage and restoring part or all of its juvenile traits [13]. Rejuvenated plants can regain juvenile morphological and physiological characteristics, thus restoring growth and reproductive vitality through the rejuvenation of old plants [1,48]. Plant metabolites are divided into primary metabolites and secondary metabolites and play crucial biological roles in important growth processes. The secondary metabolites of flavonoids and terpenoids have significant differences among the different leaf phenotypes of *C. longepaniculatum* [49]. The primary metabolites, such as amino acids, total glucose contents, and carbohydrates, were higher in YLs than MLs; in contrast, the synthesis of secondary metabolites (such as phenolics and flavonoids) was higher in MLs than YLs of sugarcane [50]. The YLs and MLs have no differences in terms of chemical compound contents, and both have very strong antioxidant activity in the agarwood species *Wikstroemia tenuiramis* [51]. In our study, 146 primary metabolites were up-regulated and 89 were down-regulated in the ML vs. RL comparison, while 54 primary metabolites were up-regulated and 38 were down-regulated in the ML vs. YL comparison. The results showed that the primary metabolite contents were higher in the juvenile tissue than the mature tissue of a *C. camphora*.

### 3.4. Gene Identification of Metabolic Pathways Related to Stress Resistance

During the long life of ancient trees, they becomes vulnerable and suffer a wide variety of biotic and abiotic stresses, including temperature, drought, waterlogging, soil nutrient deficiencies, disease, and insect pests. Trees have developed a resistance mechanism against disease and pests to protect organisms during their long-term environmental induction [52]. Leaves are the largest tissue and have direct contact with the outside environment, so leaf anatomy and structure can incur obvious changes along with tree senescence, and the leaf ultrastructure can reflect the degree of senescence in ancient trees [53]. In mature male and female leaves of ginkgo trees, miRNA-target genes are implicated in plant–pathogen interactions, plant hormone signal transduction, and flavonoid biosynthesis [54]. Therefore, complex and refined signal transduction mechanisms have gradually developed during long-term evolutionary processes to cope with various stresses [55]. The age-related changes in the studied of 15- to 667-year-old ginkgo trees indicate that the disease-resistant-associated genes remained highly expressed, and the expression levels of 15 members of the plant–pathogen interaction pathways displayed no significant differences in old trees, revealing that long-lived trees have evolved compensatory mechanisms to maintain a balance between growth and the ageing processes [56]. Leaf phenotypic, physiological and transcriptome analyses revealed that the expression levels of DEGs related to plant–pathogen interactions and plant hormone signal transduction pathways were significantly up-regulated with increasing tree ages in cutting plantlets from different years (5, 300 and 700) in regard to *Platycladus orientalis*. The results showed that old trees exhibit strong resistance but that young trees exhibit fast growth [57]. In our study, the KEGG functional enrichment analysis of DEGs showed that the plant–pathogen interaction and MAPK signaling pathway–plant pathways were important in the three ML, RL, and YL comparative analyses. A total of 56 DEGs, including WRKY33, MKK4/5, PR1, CNGCs and the RbohD transcription factor, were identified as being related to the plant−pathogen interaction pathway (ko04626). A total of 36 DEGs, including WRKY33, MKK4/5, PR1, MKK9, MYC2, CAT1, ETR/ERS, ERF1, RAN1 and RbohD, related to the MAPK signaling pathway–plant (ko04016) were identified. As an important germplasm resource, ancient *C. camphora* has been affected by the external environment and endogenous factors for hundreds of thousands of years and has developed compensatory mechanisms to respond to the growth and ageing process.

## 4. Materials and Methods

### 4.1. Plant Materials and Cutting Method

The mother tree of ancient *C. camphora* (serial number: 36012120021600006) is located in the Jingkou Township, Nanchang County, Nanchang City, Jiangxi Province (28°63′69″ N, 116°27′02″ E). According to the questionnaire, the tree was 2000-years-old and classified as first-class state protection ancient tree (Appendix A). The growth characteristics of the ancient tree is 8.8 m in height, 722 cm in chest diameter, and 15.1 m in average crown width. The trunk is hollow and rotten, only remain 2/3 bark, and many vines cover the trunk. 

Ancient *C. camphora* branches were collected in May 2020. Semi-lignified softwood spring twigs with two half leaves of 10 cm long were selected for cutting. Following the operation, after 2 min of a soaking treatment with 3000 mg/L IBA solution, the medium (nutritional:roseite:perlite = 2:1:1, *v*:*v*:*v*) humidity was maintained above 80%, the temperature was 28/22 °C (day/night), and the illumination time was 16/8 h (day/night). A sunshade net was used to avoid direct light (shading 75%). Before cutting, 0.1% carbendazim solution was used to sterilize the medium and branches once a week for repeatability. The 2-year-old cutting plantlets were planted in the Huangma Township Resources and Environment Comprehensive Experimental Base in the Jiangxi Province (Appendix A).

Healthy, free of pests and diseases, and fully expanded leaves were collected as experimental materials in September 2022. The mature leaves (MLs) (the upmost leaves from the base of current-year branches of ancient *C. camphora*), young leaves (YLs) (the downmost leaves from the base of current-year branches of ancient *C. camphora*), and rejuvenated leaves (RLs) (the current-year leaves of 2-year-old rejuvenated plantlet via cutting of ancient *C. camphora*) were collected for subsequent analysis. The fresh leaves were used to observe the phenotype and organization structure. Simultaneously, all the samples were immediately snap-frozen in liquid nitrogen and then stored at −80 °C until transcriptome and metabolome sequencing, with 3 biological replicates being for each sample.

### 4.2. Leaf Cytological Structural Observation

The middle position of the near main vein of leaves was used for structural observation. After fixation with 2.5% glutaraldehyde, a series of operations were performed: the leaves were soaked in 1% tetroxide, dehydrated with ethanol gradient, replaced with propylene oxide, impregnated, embedded and polymerized with Spurr resin, and sliced with a Leica (EMUC6) microtome at a thickness of 1 μm. After staining with 1% ferranosine solid green (TBO), the leaf cellular structure was observed and photographed under a Carl Zeiss (Primo Star) microscope.

### 4.3. Metabolome Analysis

#### 4.3.1. Plant Hormone Metabolome Sequencing and Analysis

Powdered leaf samples (100 g) were dissolved in 1170 μL acetonitrile /water/formic acid (80:19:1, *v*/*v*/*v*). After adding 10 µL ISMix-A and 20 µL ISMix-B, the mixture was immediately vortexed for 60 s and then centrifuged for 20 min (14,000 rcf, and 4 °C). The supernatant was filtered and dissolved in 200 μL acetonitrile /water/formic acid (80:19:1, *v*/*v*/*v*) and again filtered through a 0.22 μm membrane filter for subsequent ultra high-performance liquid chromatography-tandem mass spectrometry (UPLC–MS/MS) analysis. The leaf extracts were chromatographically separated by UPLC system (Agilent 1290 Infinity UPLC, Palo Alto, CA, USA). A 5500 QTRAP MSMS (Sciex, Framingham, MA, USA) was operated in both positive and negative ion modes and controlled by Analyst 1.6.3. Multiquant 3.0.3 software was used to quantify all hormones.

#### 4.3.2. Primary Metabolome Sequencing and Analysis

The leaves were vacuum freeze-dried using lyophilizer (Scientz-100F, Ningbo, China) and then ground to powder using a grinder (MM 400, Retsch, Newtown, PA, USA). An equivalent sample of 50 mg powder was added to internal standard extract with 70% methanol. The mixture was vortexed once every 30 min, each for 30 s, and repeated 6 times. After centrifugation (12,000 rpm, 3 min), the supernatant was filtered through a microporous membrane (0.22 μm pore size). The UPLC (ExionLC™ AD, Framingham, MA, USA) was used to chromatographically separate compounds of the supernatant, and the MS/MS (Applied Biosystems 4500 QTRAP, Waltham, MA, USA) was used to quantitatively analyze the hormone contents. The parameter setting of MS/MS was done according to [58]. The metabolome sequence data were aligned to the self-built MetWare DataBase (MWDB, Wuhan MetWare Biotechnology Co., Ltd., Wuhan, China). DAMs were determined by VIP ≥ 1, *p* value < 0.05 (Student’s *t* test) and |Log_2_(Fold Change)| ≥ 1.0.

### 4.4. RNA Extraction and Transcriptome Sequencing

Total RNA was extracted using an RNeasy Plant Mini Kit (Qiagen, Hilden, Germany). RNA reverse transcription was performed with a Clontech SMARTer PCR cDNA Synthesis Kit (Takara, Beijing, China). A sequencing library was constructed by screening specific fragments, and PE 150 bp sequencing was performed using an Illumina NovaSeq platform at Shanghai Personalbio Technology Co., Ltd., Shanghai, China. Raw data were filtered out for low quality and connector sequences through a quality control process. Using Bowtie2 (version 2.4.2), the clean data were matched to the reference genome of *C. camphora* (Accession number: GWHBGBX00000000), which was downloaded from https://ngdc.cncb.ac.cn/ (accessed on 10 September 2023). The gene expression level was calculated by RPKM values. Subsequently, DEGs were identified using DESeq R (version 1.24.0) software based on screening conditions (|Log_2_(Fold Change)| ≥ 1.0, *p* value < 0.05). Blast2GO 5.2 software was used for GO enrichment analysis of DEGs. KEEG enrichment analysis was performed using KOBAS 3.0 software (*p* value ≤ 0.05).

### 4.5. Quantitative Real-Time PCR (qRT–PCR) Analysis

Total RNA was extracted from leaf tissues using the TRIzol method, and the mRNA was reverse transcribed into cDNA by a Universal SYBR^®^ Green Supermix (Bio-Rad, Hercules, CA, USA) and analysed for gene expression using a SYBR Premix Ex TaqTM II (Takara) reagent on an ABI 7500 Real-time PCR instrument. The specific primers of selected DEGs were designed using Primer Premier 5, and the primer sequences are listed in Appendix A. The PCR reaction system consisted of 10 μL SYBR Premix Ex TaqTM Ⅱ, 1 μL each of forward and reverse primers (10 μmol/L), 3 μL cDNA, and 5 μL ddH_2_O, for a total volume of 20 μL. The amplification steps were as follows: 95 °C for 3 min; 95 °C for 10 s, 55 °C for 20 s, and 72 °C for 20 s, 35 cycles; 72 °C for 10 s, and 4 °C for incubation. Based on the *Ct* values of each sample, the data were analysed using the 2^−∆∆Ct^ method and a bargraph was drawn using GraphPad Prism 5. Each sample had three biological replicates, and *CcActc* (KM086738.1) was used as an internal reference gene [59,60]. 

### 4.6. Data Statistical Analysis

The data analyses were performed using IBM SPSS Statistics 19 software, and the results were presented as the mean ± standard deviation (SD). The *p* value < 0.05 between the means was considered to be significantly different, as was the *p* value < 0.01. The bargraph was drawn using GraphPad Prism 8 software. The group names of comparative analyses were defined as control vs. experimental group.

## 5. Conclusions

A comparison of ancient *C. camphora* and its cutting plantlets revealed differences in the phenotypes and tissue structures of MLs, RLs, and YLs. Pal cells were closely arranged in two layers in the RLs, but the Spo cells were loosely arranged in one layer in the MLs and YLs. The plant hormone contents analyses indicated that the IAA, iP, and iPR concentrations of the RLs were significantly higher than the MLs and YLs, but the ABA concentration of the MLs and YLs was higher than RLs (*p* < 0.05). The targeted metabolome analysis of primary metabolites revelated that a total 293 DAMs were identified in the comparison analyses (ML vs. RL, YL vs. RL, and ML vs. YL). Moreover, a total of 5241 DEGs were identified by transcriptome sequencing. According to the KEGG analysis, seven important pathways were identified, including plant hormone signal transduction (57 DEGs), plant–pathogen interaction (56 DEGs), and MAPK signaling pathway–plant (36 DEGs). KEGG enrichment conjoint analyses of DEGs and DAMs showed there were 53 common pathways in the ML vs. RL comparison, 38 common pathways in the YL vs. RL comparison, and 26 common pathways in the ML vs. YL comparison. A total of 16 commonly enriched KEGG pathways were identified in the three comparison analyses.

## Figures and Tables

**Figure 1 ijms-25-07664-f001:**
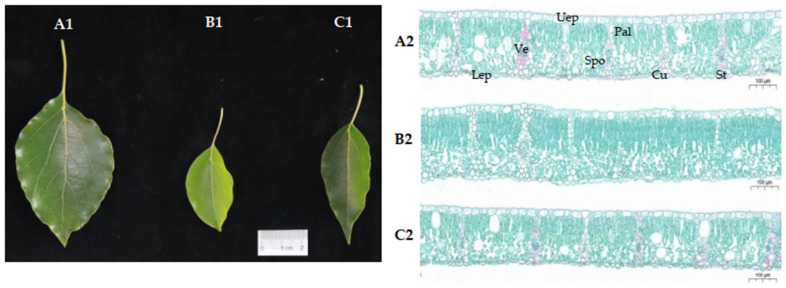
Leaf phenotype and tissue structure in the MLs, RLs, and YLs of a 2000-year-old ancient *C. camphora* and its 2-year-old cutting plantlets (A: ML, B: RL, C: YL; Subscript 1: leaf phenotype, Subscript 2: tissue structure). (Uep: upper epidermis, Pal: palisade tissue, Spo: spongy tissue, Ve: vessel, Lep: lower epidermis, St: stomata, Cu: cuticle).

**Figure 2 ijms-25-07664-f002:**
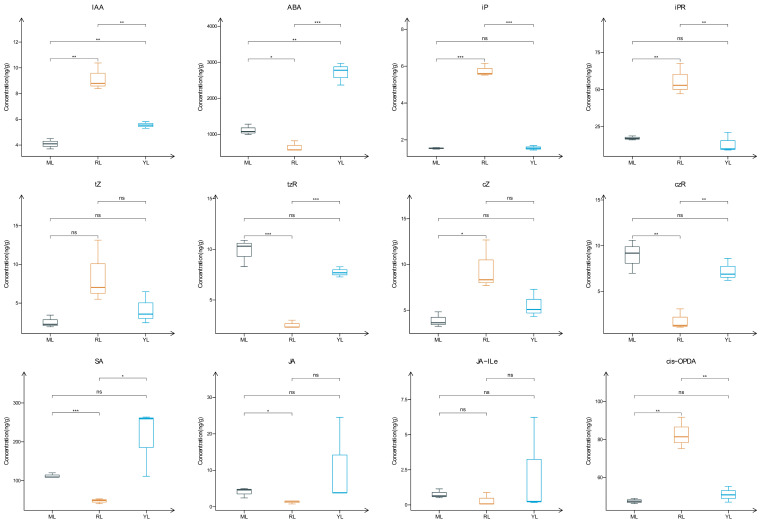
Boxplots of 12 plant hormone concentrations in MLs, RLs and YLs. Statistically significant differences were determined using Student’s *t*-test; the significance symbols ***, ** and * represent *p* values of 0.001, 0.01 and 0.05, respectively; ns represents not significant. (IAA (indole-3-acetic acid), ABA (abscisic acid), iP (isopentenyladenine), iPR (isopentenyladenosine), tZ (trans-zeatin), tzR (trans-zeatin riboside), cZ (cis-zeatin), czR (cis-zeatin riboside), SA (salicylic acid), JA (jasmonic acid), JA-ILe (jasmonoyl-isoleucine), and cis-OPDA (cis-12-oxo-phytodienoic acid)).

**Figure 3 ijms-25-07664-f003:**
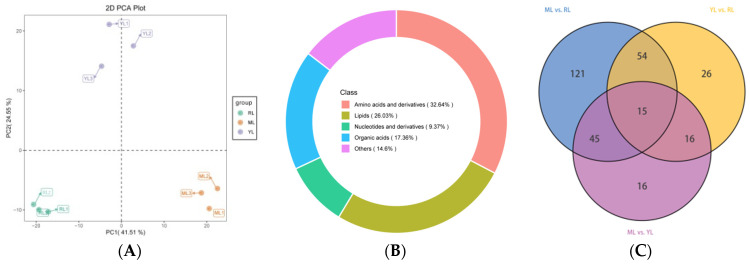
Metabolomic data analysis of YLs, MLs and RLs. Principal component analysis (PCA) (**A**), metabolite classes (**B**), and Venn diagram of DAMs (**C**).

**Figure 4 ijms-25-07664-f004:**
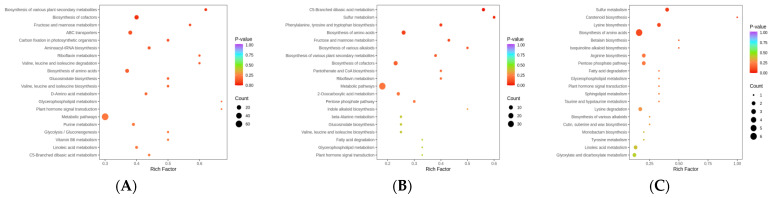
KEGG enrichment map of DAMs ((**A**) ML vs. RL; (**B**) YL vs. RL; (**C**) ML vs. YL).

**Figure 5 ijms-25-07664-f005:**
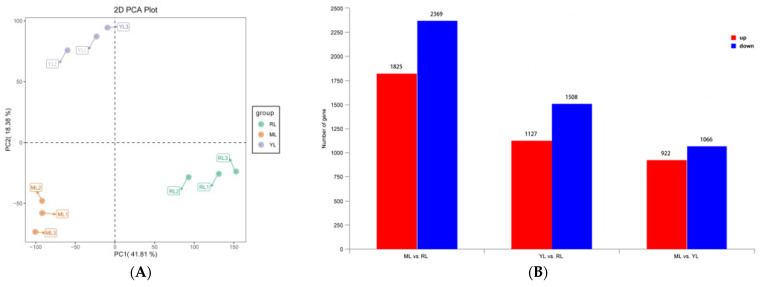
The DEG summary results between MLs, RLs, and YLs. Principal component analysis (PCA) (**A**), histogram of up- and down-regulated DEG numbers (**B**), heat map of cluster analysis (**C**), and Venn diagram of the DEGs (**D**).

**Figure 6 ijms-25-07664-f006:**
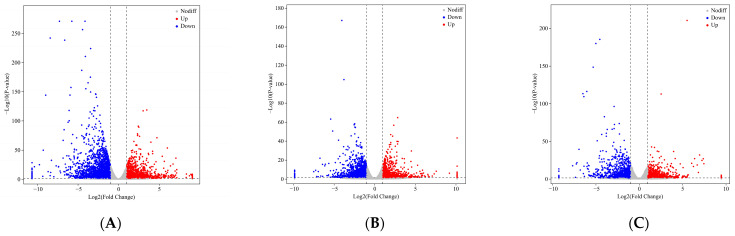
Volcano plots of DEGs (**A**–**C**) and scatter plots of the top 20 enriched KEGG pathways (**D**–**F**) ((**A**,**D**): ML vs. RL; (**B**,**E**): YL vs. RL; (**C**,**F**): ML vs. YL).

**Figure 7 ijms-25-07664-f007:**
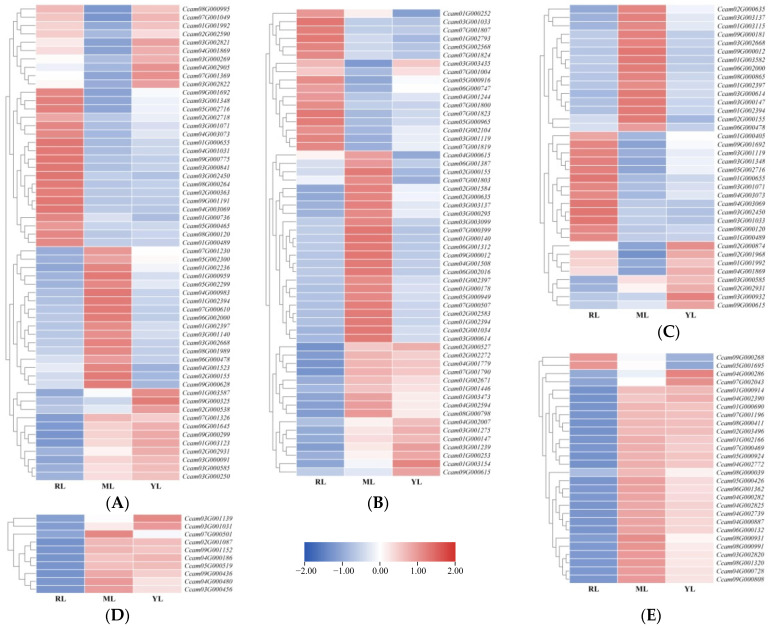
DEGs in the plant hormone signal transduction (**A**), plant–pathogen interaction (**B**), MAPK signaling pathway–plant (**C**), photosynthesis (**D**), and photosynthesis-antenna proteins (**E**) pathways between the RL, ML, and YL groups.

**Figure 8 ijms-25-07664-f008:**
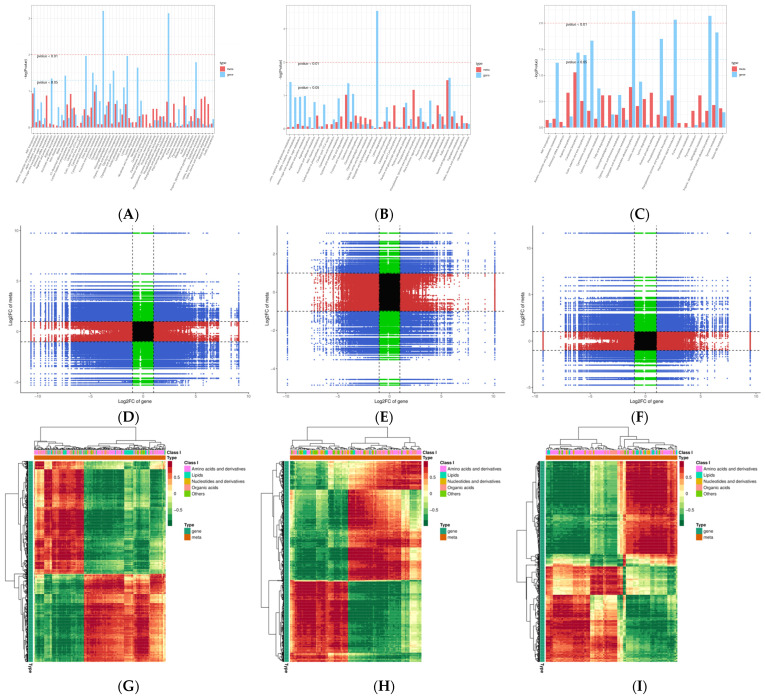
Conjoint analyses of DEGs and DAMs. KEGG enrichment (**A**–**C**), nine quadrant diagram (**D**–**F**), and correlation coefficient cluster heatmap (**G**–**I**). ((**A**,**D**,**G**): ML vs. RL; (**B**,**E**,**H**): YL vs. RL; (**C**,**F**,**I**): ML vs. YL).

**Figure 9 ijms-25-07664-f009:**
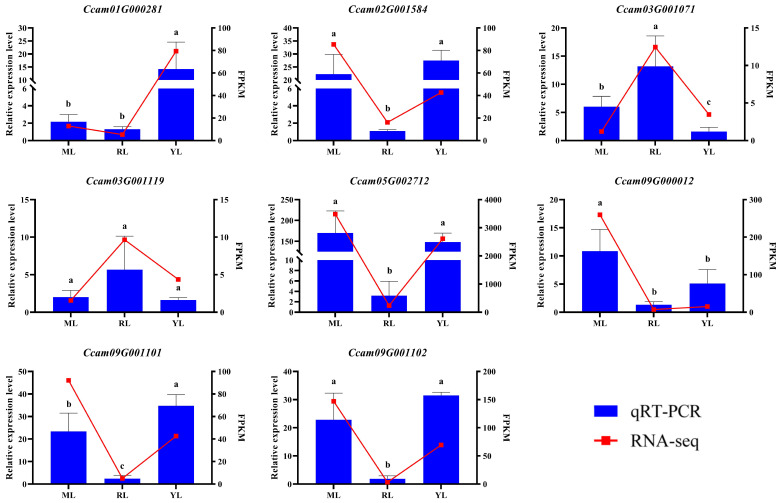
qRT-PCR validation of DEGs from RNA-seq. Error bars indicate the standard deviation (SD). Small letters (a, b and c) represent statistically significant difference (*p* value < 0.05) using a Student’s *t*-test.

**Table 1 ijms-25-07664-t001:** Statistical table of the number of DAMs in the ML, RL, and YL groups.

Group Name	All Significant Difference	Down-Regulated	Up-Regulated
ML vs. RL	235	89	146
YL vs. RL	111	62	49
ML vs. YL	92	38	54

## Data Availability

The supplementary data can be downloaded from the Appendix A. The raw RNA-Seq transcriptome data have been deposited in the NCBI Sequence Read Archive (SRA) under accession number PRJNA1110378.

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
