# Peer review of "Transcriptome and Metabolome Analyses of Leaves from Cutting Rejuvenation of Ancient Cinnamomum camphora"

_ijms, 2024, doi:10.3390/ijms25147664_

Round 1

Reviewer 1 Report

Comments and Suggestions for Authors

The manuscript by Liu et al performed the transcriptome and metabolome analysis of leaves from cutting rejuvenation of ancient Cinnamomum camphora. A total of 293 differentially accumulated metabolites and 5241 differentially expressed genes (DEGs) were identified. Integration analyses of cytological, hormone, metabolome and transcriptome may provide research basis of rejuvenation regulatory mechanism of ancient C. camphora.

Specific comments:

1.     The main concern was that although the manuscript revealed a lot of differences in so many aspects, what’s the main point of these results? The author should clarify them.

2.     All the results were based on dry lab analysis, did the author performed some wet-experiments ?

3.     Fig 1, what did A1 and A2 represent respectively?

4.     Some spelling errors.

Comments on the Quality of English Language

 Moderate editing of English language required

Author Response

Point-by-point response to Comments and Suggestions for Authors

Comments 1: The main concern was that although the manuscript revealed a lot of differences in so many aspects, what’s the main point of these results? The author should clarify them.

Response 1: The results reveal the have significant difference in DEGs and DAMs on ancient C. camphora and its cutting plantlets

Comments 2: All the results were based on dry lab analysis, did the author performed some wet-experiments ?

Response 2: Leaves tissue structure and qRT‒PCR was consider as wet-experiments.

Comments 3: Fig 1, what did A1 and A2 represent respectively?

Response 3: I have modified A1 and A2 in the line 217-218 of Figure 1.

Comments 4: Some spelling errors.

Response 4: I have modified some spelling errors in the in the full revise manuscript.

Comments 5: Moderate editing of English language required

Response 5: I have modified some sentences in the in the uploaded revise manuscript.

Reviewer 2 Report

Comments and Suggestions for Authors

This manuscript reports a comprehensive study on the transcriptome and metabolome analyses of leaves from the cutting rejuvenation of ancient Cinnamomum camphora. Overall, the results and data may contribute valuable insights into the rejuvenation mechanisms of ancient trees. However, there are some major issues that need to be addressed:

The introduction and discussion of the paper are not well written, with weak logical connections between sentences and contexts, requiring careful revision.

The definition of rejuvenation is incorrect. Line 9 states, "Rejuvenation refers to the transition from the state of juvenile to mature." Lines 29–32 state, "from juvenile to adult development stage." This wording is wrong; rejuvenation refers to the transition from the adult to juvenile stage, not from juvenile to adult. Carefully check the “rejuvenation” throughout the manuscript to avoid such a mistake.

Line 42: Because ancient trees are widespread in many countries worldwide, why only mention "the Chinese nation" and not other countries?

Lines 53-54, "However, the rejuvenation mechanisms have not been characterized in ancient C. camphora." The authors should first introduce how cuttings can rejuvenate the aging branches of ancient trees before stating that cutting rejuvenation mechanisms have not been characterized in ancient C. camphora.

Line 140, "a 2000-year-old ancient C. camphora." How did you determine the age of this tree to be 2000 years old?

All "cutting seeding" should be changed to "cuttings." The authors may refer to "seedlings," not "seeding." However, seedlings are young plants germinated from seeds, not from cuttings.

While the study integrated transcriptomic and metabolomic data, a more thorough analysis of the interactions between differentially expressed genes (DEGs) and differentially accumulated metabolites (DAMs) would provide deeper insights into the regulatory mechanisms of rejuvenation in ancient C. camphora.

In the discussion section, the authors should discuss how observed differences in palisade tissue cell layers and hormonal concentrations contribute to the rejuvenation process.

Plant secondary metabolites play essential roles in the stress tolerance of ancient trees (Tree Physiology, 2024, 44, tpae002). Although this study did not identify differential secondary metabolites, the enrichment of DEGs results in phenylpropanoid biosynthesis pathways. C. camphora is rich in secondary metabolites, such as phenylpropanoids, especially in the leaves. Therefore, the analysis of this aspect should be included in the results and discussion sections to cover more findings of the rejuvenated process, thereby improving the quality of the study.

Figure 1. A1, B1, and C1 should include a scale bar.

The manuscript needs thorough proofreading to correct grammatical errors and ensure academic correctness.

Author Response

Point-by-point response to Comments and Suggestions for Authors

Comments 1: The introduction and discussion of the paper are not well written, with weak logical connections between sentences and contexts, requiring careful revision.

Response 1: The introduction and discussion have carefully revised in many Sentences and paragraphs in the uploaded revise manuscript.

Comments 2: The definition of rejuvenation is incorrect. Line 9 states, "Rejuvenation refers to the transition from the state of juvenile to mature." Lines 29–32 state, "from juvenile to adult development stage." This wording is wrong; rejuvenation refers to the transition from the adult to juvenile stage, not from juvenile to adult. Carefully check the “rejuvenation” throughout the manuscript to avoid such a mistake.

Response 2: Thank you for pointing this out. I agree with this comment. Therefore, I have modified the sentence in the uploaded revise manuscript.

Comments 3: Line 42: Because ancient trees are widespread in many countries worldwide, why only mention "the Chinese nation" and not other countries?

Response 3: I agree with this comment. I have deleted the sentence of Chinese nation in the line 41 of uploaded revise manuscript.

Comments 4: Lines 53-54, "However, the rejuvenation mechanisms have not been characterized in ancient C. camphora." The authors should first introduce how cuttings can rejuvenate the aging branches of ancient trees before stating that cutting rejuvenation mechanisms have not been characterized in ancient C. camphora.

Response 4: I agree with this comment. I have modified the sentence (how can juvenile characteristics through cutting, have not been characterized in ancient C. camphora) in the line 63 of revise manuscript.

Comments 5: Line 140, "a 2000-year-old ancient C. camphora." How did you determine the age of this tree to be 2000 years old?

Response5: According to the questionnaire, there has a small signboard in the Figure S6A. The ancient C. camphora has been recorded 2000-year-old in the signboard.

Comments 6: All "cutting seeding" should be changed to "cuttings." The authors may refer to "seedlings," not "seeding." However, seedlings are young plants germinated from seeds, not from cuttings.

Response 6: I agree with this comment. I have modified the sentence of cutting seeding to cutting plantlets in the uploaded revise manuscript.

Comments 7: While the study integrated transcriptomic and metabolomic data, a more thorough analysis of the interactions between differentially expressed genes (DEGs) and differentially accumulated metabolites (DAMs) would provide deeper insights into the regulatory mechanisms of rejuvenation in ancient C. camphora.

Response 7: In the line 365, 2.5. The sequence data conjoint analysis of DEGs and DAMs,

Comments 8: In the discussion section, the authors should discuss how observed differences in palisade tissue cell layers and hormonal concentrations contribute to the rejuvenation process.

Response 8: I agree with this comment. I have increase the function of palisade tissue cell layers in the line 456-457. I have increase the hormonal function in the line 473-474.

Comments 9: Plant secondary metabolites play essential roles in the stress tolerance of ancient trees (Tree Physiology, 2024, 44, tpae002). Although this study did not identify differential secondary metabolites, the enrichment of DEGs results in phenylpropanoid biosynthesis pathways. C. camphora is rich in secondary metabolites, such as phenylpropanoids, especially in the leaves. Therefore, the analysis of this aspect should be included in the results and discussion sections to cover more findings of the rejuvenated process, thereby improving the quality of the study.

Response 9: From Figure 6, the DEGs number involved of phenylpropanoid biosynthesis pathways were behind in the top 20 enriched KEGG pathways.

Comments 10: Figure 1. A1, B1, and C1 should include a scale bar.

Response 10: I agree with this comment. The Figure 1 has increased a scale bar.

Comments 11: The manuscript needs thorough proofreading to correct grammatical errors and ensure academic correctness.

Response 11: We have proofreading grammatical errors in uploaded revise manuscript.

Reviewer 3 Report

Comments and Suggestions for Authors

Dear Authors,

The manuscript ijms-304834 entitled "Transcriptome and metabolome analyses of leaves from cutting rejuvenation of ancient Cinnamomum camphora" focuses on the morphological, biochemical and molecular comparisons among leaves from an ancient C camphora tree and those of a two-year-old plant obtained from a cutting of the same tree.

The analysis provides data about the hormone content, transcriptome and metabolome analyses performed in three types of leaves: young (YL) and mature (ML) leaves from the old tree and the called rejuvenated leaves (RL) harvested from the cutting-derived plant.

 The authors found anatomical, hormone content differences between RL and leaves from the ancient tree (YL and ML), as well as differentially accumulated metabolites (DAM) in the different groups. They also found differentially expressed genes (DEGs) and common enriched pathways related to the plant hormone signal transduction, plant-pathogen interaction and MAP signalling pathways in the three types of leaves.

Although the study provided interesting data, and fits into the scope of the journal, the manuscript presents several shortcomings and needs important modifications before publication.           

1-      My main concern is related to the design of the study. Authors compared young and mature leaves harvested from the same old tree with leaves (RL) taken from a two-year-old rooted cutting derived from crown branches of the same tree. However, as we can see in Fig. S6, the 2-yeard-old plant has leaves of different sizes, maturation and physiological stages along the new branches, ie, younger leaves in the uppermost part of the new branches and older leaves at the base of the branches. The authors infer that leaves derived from the 2-year-old cutting are rejuvenated, however in my opinion they are leaves from a young plant.

In the case of rejuvenated Ginkgo, which is mentioned by the authors, the new shoots developed along the trunk are derived from latent/ adventitious buds which are ontogenetically young and exhibit juvenile characteristics (Jang et al., 2021, Lu et al., 2022).

2-      The language style can be improved because sometime confuse the reader

3-      The authors should clearly state the aim of this study and the hypothesis. Why they carried out the study? Is it related to rejuvenation, to vegetative propagation or just about changes involved through plant development??

4-      Introduction should be rewritten and shortened. Some paragraphs are a list of references which in some cases are not in accordance with what has been published. For example, the reference 6 in lane 100. The sentence from line 98-100 does not reflect what the authors state “ What is the meaning of “the expressed genes… were closely related to leaves of resprouters and old branches in ginkgo”.? According to the authors, genes related to photosynthesis, auxin signaling, stress-associated genes were upregulated in leaves of resprouters.

Lines 112-113-“The YL of Hevea braisileinsis are more susceptible to biotic and abiotic stress than ML and 67,8 % DEGs were expressed in ML” . This a very simple interpretation of the results published by Fang et al.  (2016). They found that 67,8 %of DEGs involved in leaf development were identified during the transition to leaf maturation. The different expression patterns of genes involved in lignin biosynthesis or cyanogenic activity in juvenile and mature leaves correlate with changes in disease resistance of leaves.

5-      Figures 2 and 4 should be enlarged

6-      In line 102:” it can significantly promote rejuvenation” can be removed as it is somehow redundant, as they initiated the sentence: Trunk truncation is an important rejuvenation method.

7-      In line 159: The last sentence should be rewritten

8-      Which statistical analysis has been used for checking the significance of qRT-PCR data? (figure 9). If done, why it is not represented in the graph?

9-      Discussion can be shortened and summarized

10-  Conclusions are just a data summary

Author Response

Point-by-point response to Comments and Suggestions for Authors

Comments 1: My main concern is related to the design of the study. Authors compared young and mature leaves harvested from the same old tree with leaves (RL) taken from a two-year-old rooted cutting derived from crown branches of the same tree. However, as we can see in Fig. S6, the 2-yeard-old plant has leaves of different sizes, maturation and physiological stages along the new branches, ie, younger leaves in the uppermost part of the new branches and older leaves at the base of the branches. The authors infer that leaves derived from the 2-year-old cutting are rejuvenated, however in my opinion they are leaves from a young plant.

Response 1: The Figure S6B representative the growth season From March to April , we take samples in the September, so we upload another Figure of September to replace Figure S6B in the Supplementary Files.

Comments 2:  The language style can be improved because sometime confuse the reader.

Response 2: The language style has been modified in the upload revise manuscript.

Comments 3: The authors should clearly state the aim of this study and the hypothesis. Why they carried out the study? Is it related to rejuvenation, to vegetative propagation or just about changes involved through plant development?

Response 3: In this study, the hormones content, primary metabolites and gene expression levels have significant change in the cutting rejuvenation of ancient Cinnamomum camphora.

Comments 4: Introduction should be rewritten and shortened. Some paragraphs are a list of references which in some cases are not in accordance with what has been published. For example, the reference 6 in lane 100. The sentence from line 98-100 does not reflect what the authors state “ What is the meaning of “the expressed genes… were closely related to leaves of resprouters and old branches in ginkgo”.? According to the authors, genes related to photosynthesis, auxin signaling, stress-associated genes were upregulated in leaves of resprouters.

Lines 112-113-“The YL of Hevea braisileinsis are more susceptible to biotic and abiotic stress than ML and 67,8 % DEGs were expressed in ML” . This a very simple interpretation of the results published by Fang et al.  (2016). They found that 67,8 %of DEGs involved in leaf development were identified during the transition to leaf maturation. The different expression patterns of genes involved in lignin biosynthesis or cyanogenic activity in juvenile and mature leaves correlate with changes in disease resistance of leaves.

Response 4: Thank you for pointing this out. I agree with this comment. Therefore, I have modified the sentence in the line 109 of revise manuscript. I have modified the sentence in the line 148-151 of revise manuscript.

Comments 5:  Figures 2 and 4 should be enlarged.

Response5: The Figures 2 and 4 have been enlarged in the revise manuscript.

Comments 6:  In line 102:” it can significantly promote rejuvenation” can be removed as it is somehow redundant, as they initiated the sentence: Trunk truncation is an important rejuvenation method.

Response6: I agree with this comment. I have been removed the sentence of significantly promote rejuvenation in the line 139.

Comments 7: In line 159: The last sentence should be rewritten.

Response 7: I agree with this comment. The last sentence has been modified in the line 211 of revise manuscript.

Comments 8: Which statistical analysis has been used for checking the significance of qRT-PCR data? (figure 9). If done, why it is not represented in the graph?

Response 8: I agree with this comment. The Figure 9 has been increase statistically significant difference in the revise manuscript.

Comments 9:  Discussion can be shortened and summarized.

Response 9: I agree with this comment. Some sentences have been deleted in the discussion in the upload revise manuscript.

Comments 10:  Conclusions are just a data summary

Response 10: Conclusions have been some modified.

Reviewer 4 Report

Comments and Suggestions for Authors

The manuscript with the title “Transcriptome and metabolome analyses of leaves from cutting rejuvenation of ancient Cinnamomum camphora”, is novel and relevant study on the rejuvenation regulatory mechanism of C. camphora. The study focused on comparative analysis of an ancient tree and young cutting to highlight mechanisms. Ancient trees are valuable for biodiversity and living repositories of biologic information. Few studies document their physiological and molecular mechanisms responsible for their rejuvenation and consequently longevity. Given the low and decreasing number of ancient living specimens their scientific study might be a last chance to untangle and document key aspects of their biology. I consider this constitutes the unquestionable value of the study, and the contribution is of particular interest from this regard.

The introduction is very well written and relevant literature is summarized.

Figure 2 components are small and could be enlarged. I suggest caption to also include the test based on which the significance was determined.

Figure 3 a – I suggest to insert as separate, larger figure the PCA projection.

Figures 4 and 6 are nearly impossible to read. I suggest authors to somehow find a way to present them larger.

I suggest to renounce to abbreviations in Abstract and Conclusions (for the experimental variants) to facilitate the readers to find faster what they seek and to ensure these sections are stand-alone explanatory.

Best regards.

Comments on the Quality of English Language

fine/minor English style and grammar improvements are recommended.

Author Response

Point-by-point response to Comments and Suggestions for Authors

Comments 1: Figure 2 components are small and could be enlarged. I suggest caption to also include the test based on which the significance was determined.

Response 1: The Figure 2 has been enlarged, and increase statistically significant difference in the line 245-246.

Comments 2: Figure 3 a – I suggest to insert as separate, larger figure the PCA projection.

Response 2: The Figure 3a has been enlarged.

Comments 3: Figures 4 and 6 are nearly impossible to read. I suggest authors to somehow find a way to present them larger.

Response 3: The Figures 4 and 6 has been enlarged.

Comments 4: I suggest to renounce to abbreviations in Abstract and Conclusions (for the experimental variants) to facilitate the readers to find faster what they seek and to ensure these sections are stand-alone explanatory.

Response 4: The terminology abbreviations have the full name in the Abstract. I consider can retain the abbreviations in the Conclusions.

Comments 5: fine/minor English style and grammar improvements are recommended.

Response 5: The grammar has been modified in the upload revise manuscript.

Round 2

Reviewer 3 Report

Comments and Suggestions for Authors

Dear Authors,

I revised carefully the new version and some aspects have been improved. However there are some "tipo" and grammatical errors  which have not been addressed. I marked some of them (highlighted) in the pdf version.

The conclusions section has not been  modified . I would suggest to the authors that they rework their conclusions so that they are not merely an account of results.

Author Response

Comments 1: I revised carefully the new version and some aspects have been improved. However there are some "tipo" and grammatical errors which have not been addressed. I marked some of them (highlighted) in the pdf version.

Response 1: I agree with this comment. I have modified some "tipo" and grammatical errors in the uploaded revise manuscript.

Comments 2: The conclusions section has not been  modified . I would suggest to the authors that they rework their conclusions so that they are not merely an account of results.

Response 2: I agree with this comment. I have modified the conclusions in the uploaded revise manuscript.